# Has Inositol Played Any Role in the Origin of Life?

**DOI:** 10.3390/life7020024

**Published:** 2017-06-05

**Authors:** Adolfo Saiardi

**Affiliations:** Medical Research Council Laboratory for Molecular Cell Biology, University College London, London WC1E 6BT, UK; dmcbado@ucl.ac.uk; Tel.: +44-20-7679-7254

**Keywords:** inositol, pyrophosphate, hypothesis, metabolism, phosphorylation, sugar

## Abstract

Phosphorus, as phosphate, plays a paramount role in biology. Since phosphate transfer reactions are an integral part of contemporary life, phosphate may have been incorporated into the initial molecules at the very beginning. To facilitate the studies into early phosphate utilization, we should look retrospectively to phosphate-rich molecules present in today’s cells. Overlooked by origin of life studies until now, inositol and the inositol phosphates, of which some species possess more phosphate groups that carbon atoms, represent ideal molecules to consider in this context. The current sophisticated association of inositol with phosphate, and the roles that some inositol phosphates play in regulating cellular phosphate homeostasis, intriguingly suggest that inositol might have played some role in the prebiotic process of phosphate exploitation. Inositol can be synthesized abiotically and, unlike glucose or ribose, is chemically stable. This stability makes inositol the ideal candidate for the earliest organophosphate molecules, as primitive inositol phosphates. I also present arguments suggesting roles for some inositol phosphates in early chemical evolution events. Finally, the possible prebiotic synthesis of inositol pyrophosphates could have generated high-energy molecules to be utilized in primitive trans-phosphorylating processes.

## 1. Introduction

The element phosphorus, in its phosphate form (PO_4_^−3^), plays a paramount role in biology. Phosphate’s negative charges at physiological pH, its tetrahedral molecular geometry, and its ability to coordinate hydrogen bonds through its oxygen groups are characteristics that have made this molecule essential in living organisms. It plays crucial structural roles in the backbone of nucleic acids or organising membranes in phospholipids. Attachment of phosphate groups to proteins is one of the principal regulatory mechanisms in biology. More importantly, the energy stored in phosphoanhydride bonds drives the majority of biological processes. Life as we know it depends on phosphate [1,2]. Therefore, any endeavour theorising processes that led to the origin of life should consider phosphorus chemistry, and its initial exploitation by prebiotic processes. This is somewhat controversial, since the iron-sulphur hypothesis [3,4], which envisages the reduction of metal-sulphide minerals for the origin of metabolism, and does not foresee any role for phosphate, has gained substantial attention since the 1990s. However, it is undeniable that evolution has exploited phosphorus and not sulphur as the energetic currency. The central molecule of today’s intermediate metabolism is ATP, adenosine triphosphate, and not a hypothetical “adenosine trisulphate”.

There are some concerns on how the exploitation of phosphorus would have been achieved, as under the reducing conditions known to have existed on early Earth phosphorus would have been sequestered in insoluble minerals [5,6]. It is possible that the initial phosphorus, which later became organophosphate, instead originated from the meteoritic bombardment of Earth, as recent work on the meteoritic mineral schreibersite has indicated [7,8]. Furthermore, urea-based solvents, compatible with early Earth conditions, are known to enhance the mobilization of phosphate from minerals, as well as the abiotic phosphorylation of organic molecules, especially in the presence of magnesium [9]. Intriguingly, magnesium is often the essential cofactor in today’s cellular biochemical processes involving phosphate groups [10]. This suggests that we should look closely at several aspects of current phosphate biology: knowing how phosphate homeostasis is managed now could give insights into prebiotic phosphate exploitation. Another abundant bivalent cation present in early Earth’s oceans was calcium. This mineral has the unfortunate predisposition to precipitation in the presence of phosphate, something not compatible with the biochemical processes of life [11,12]. Extraordinarily, phosphate and calcium homeostasis in today’s eukaryotic cells are regulated by different members of the same class of molecules, the inositol phosphates [13,14]. This leads to the question: can the biology of inositol phosphates give insight into the prebiotic processes of phosphate exploitation?

## 2. Inositol and Inositol Phosphates in Biology Today

Today’s cellular biochemistry has been shaped by billions of years of chemical and biological evolution. Pathways that could have been involved in the origin of life, are likely to be present today, but perhaps partially shielded by more “elegant” and complex roles that evolution has provided. A sugar, like inositol, could certainly fit this description as it has been exploited by evolution to create a vast array of signalling molecules present in eukaryotic cells today. Various characteristics of this cyclitol mean that it could have played some role in prebiotic times. The fact that inositol has a close relationship with phosphate particularly advocates for an initial role of this sugar in prebiotic phosphate exploitation [14].

Here, I will underline some of the modern biology of inositol that will be useful to appreciate the roles that this simple and almost ubiquitous sugar alcohol might have played in the origin of life. While the formose reaction has been shown to synthesize inositol abiologically (see below), its synthesis in cells today involves the irreversible isomerisation of glucose-6-P into inositol-3-P, a far more stable sugar (Figure 1A). This conversion is carried out by most Archaea, some bacteria, and virtually all eukaryote life forms, meaning inositol is commonly present in two out of three kingdoms of life [15,16]. Perhaps one of the most important but often overlooked characteristics of inositol is its metabolic and chemical stability. Unlike its precursor glucose (Figure 1B), whose carbon backbone can be used to generate all the other organic molecules, the carbons of inositol remain as inositol (the structure of *myo-*inositol, the most common isomer, is presented in Figure 1C). This makes inositol an ideal osmolyte, a property utilised by Archaea, as well as mammalian neurons [15,17,18]. Inositol also provides a metabolically inert and versatile canvas that can be decorated with phosphate groups (Figure 2A–C). Eukaryotic cells in particular have exploited this to generate a multifaceted array of phosphorylated signalling molecules [19,20]. The six-carbon ring can be considered a six-bit code, with the potential to encode 64 unique species [21]. These phosphoester bond-containing molecules, the inositol phosphates, are ubiquitously present in eukaryotic cells. Higher phosphorylation complexity derives from the ability of carbon atoms to host more than one phosphate, generating the so-called inositol pyrophosphates (Figure 2C), with highly energetic phosphoanhydride bonds [22,23,24]. Further variety is provided by the phosphorylated inositol lipids, referred to as phosphoinositides, that have a phosphodiester moiety bridging the inositol to the lipid [25].

This varied family of molecules first attracted attention after the demonstration that the specific receptor-dependent production of one member, I(1,4,5)P_3_, from the lipid phosphoinositide PI(4,5)P_2_ mobilises calcium from intracellular stores [32]. The intense research that followed revealed the ubiquitous distribution and fundamental importance of lipid-bound inositol phosphates in cell biology. This included elucidating their role in defining membrane identity [33], and the importance of the lipid PI(3,4,5)P_3_ in controlling growth and differentiation signalling pathways [34]. The well-characterised I(1,4,5)P_3_-regulated calcium signalling paradigm, as well as PI(3,4,5)P_3_ signalling cascades, conceivably represent the pinnacle of the functions that evolution has provided to inositol phosphates. Indeed, these two signalling paradigms are mainly characteristic of higher metazoans, with several eukaryote clades, beside bacteria and Archaea, not having either.

While only seven phosphoinositide species are known, of the more than 30 cytosolic inositol phosphates identified so far, very few of them have universally-accepted functions [19]. This lack of knowledge is primarily due to the complexity of the phosphorylation patterns, and the lack of easy experimental tools to study them. It is possible that many inositol phosphates are merely metabolic intermediates for more complex species. Even for the inositol pyrophosphates (Figure 2C), to which many specific functions have been attributed, debate continues as to their fundamental role [35,36]. However, converging lines of evidence have recently indicated that these molecules regulate cellular phosphate homeostasis [13,37,38]. Inositol pyrophosphates are evolutionarily ancient molecules: the enzymes for synthesising them are omnipresent in eukaryote genomes [39,40]. This has led to speculation that their ability to control phosphate homeostasis represents the original/initial function of all inositol phosphates [14]. One appealing possibility is that this primitive inositol phosphate/pyrophosphate signalling, by modulating cellular phosphate, helped establish a primitive calcium signalling. The regulated secretion of calcium from prokaryotic cells and eukaryotic cytosol is a fundamental characteristic of life, since calcium precipitates both organic and inorganic phosphate forming insoluble salts [12]. The complexity of this system could have then evolved in parallel with eukaryotic cell sophistication, ending up with today’s I(1,4,5)P_3_-regulated calcium signalling paradigm.

Inositol pyrophosphates control cellular phosphate homeostasis at multiple levels in eukaryotes. Some of this control relies on their ability to regulate the metabolism of inorganic polyphosphate (polyP), a linear polymer of phosphate groups joined by phosphoanhydride bonds [41,42]. At least in yeast, inositol pyrophosphates tightly regulate the cellular presence of polyP [37,38,43]. Though recent research has demonstrated the importance of polyP in modern cell physiology [44,45,46], and for many years polyP was regarded as a “molecular fossil” [47,48]. This polymer is indeed thought to have been present on early Earth, perhaps synthesised prebiotically by volcanic activity [49]. It has been proposed that polyP could have acted as the phosphate donor for prebiotic phosphotransfer reactions [47]. Another likely early-Earth molecule that could have participated in phosphotransfer reactions is pyrophosphate (essentially polyP of chain length two). However, it is remarkable that the inositol pyrophosphates themselves have the ability to participate in phosphotransfer reactions. More precisely, these molecules are able to transfer the β-phosphate to a pre-phosphorylated serine, generating a pyrophospho-serine species [50,51].

With the above arguments in mind, and considering the intimate relationship between phosphate and inositol in existing cells, it is tempting to propose several roles for inositol during prebiotic chemistry.

## 3. Inositol Prebiotic Synthesis

To hypothesise that inositol might have played a role in the origin of life, one must first accept that its synthesis must have occurred prebiotically. Cyanide has attracted interest from chemists working on the prebiotic synthesis of organic molecules, since it is the most abundant carbon-containing compound in the interstellar medium [52]. Cyanide reacts with water to form formamide, which has also been detected in space. The abiotic chemistry of both molecules, or of their derivatives urea, ammonium formate and formamide, can give rise to nucleic acid precursors [53]. Formamide appears to be a valid precursor for the synthesis of nucleosides [54] but, importantly, also acts to catalyse their phosphorylation [53,55]. Formamide abiotic chemistry is complex and varied: the major classes of organic molecules, including nucleobases, nucleosides, carboxylic acids, amino acids, and sugars, can be synthesised by bombarding formamide with high-energy protons, in the presence of meteorite powder as a catalyst [56]. The synthesis of sugars from formamide most likely involves its conversion to formaldehyde and the well-studied formose reaction [57], an aldolic-like condensation [58]. The formose reaction could also have been responsible for prebiotic synthesis of inositol [59]. Meteorite-catalysed synthesis of inositol from formamide under proton irradiation has, in fact, been recently observed [56]. Specifically, catalysis by the stony-iron NWA 4482 meteorite resulted in 2.88 µg of inositol created per ml of formamide, making it the most abundant sugar synthesized. This important result should not come as a total surprise since inositol has been previously identified in the carbonaceous Murchison meteorite [60]. These two pieces of evidence indicate that prebiotic synthesis of inositol is not only possible from formamide/formaldehyde [56], but has actually occurred in a meteorite [60].

How the initial phosphate fixation into organic molecules occurred is a highly debated argument [6]. Phosphate in prebiotic Earth was unlikely to be freely available, as its affinity for bivalent cations would cause its precipitation from solution. This problem seems unsurmountable to many scientists who envisage an initial phosphate-free metabolism [4] mainly founded on iron and sulphur chemistry [3,61]. While these arguments are valid, current cellular metabolism is actually based on phosphate. The formation of organophosphate molecules must, therefore, have occurred in a very early phase in the process that led to the origin of life. In the past decade, several lines of evidence have made the prebiotic synthesis of organophosphate molecules plausible. One of the most credible routes involves meteorites providing the phosphide-containing mineral schreibersite (see below). The early Earth-plausible eutectic solvent urea/ammonium formate/water, in the presence of magnesium, has also been shown to mobilize phosphate from insoluble minerals, markedly increasing the phosphorylation of organic molecules [9].

Research into the origin of life has, so far, given no attention to inositol and/or inositol phosphates, therefore, the absence of such molecules from the literature is not a surprise. If inositol’s prebiotic existence is proven, what about its phosphorylation to inositol phosphates? Since nucleosides [9,53,62] and glycerol [9], both possessing three hydroxyl groups, have been phosphorylated to generate several phosphate ester moieties using abiotic chemistry, it is reasonable to assume that the more stable inositol with its six hydroxyl groups could similarly act as an abiotic phosphate group acceptor. Therefore, the prebiotic formation of inositol phosphates, although not yet demonstrated, is certainly plausible.

## 4. Inositol and Inositol Phosphates on Early Earth

### 4.1. Inositol and the Initial Organophosphate Molecule 

If phosphate-based chemistry was incorporated in the most ancient biochemical machineries [61], the first question to address is: where did this phosphorus come from? The most intriguing observation concerns schreibersite, (Fe,Ni)_3_P. This mineral is able to phosphorylate organic compounds [8,62], and by releasing phosphide, can generate an array of P-oxyacids that, through a series of transformations, can also produce the pyrophosphate [63]. Until recently, experiments testing mineral phosphorylation of organic molecules were performed in the presence of anhydrous solvents [53], or without the presence of water [64], conditions unlikely to be present on early Earth. Recently, protocols have been established that enable a good degree of phosphorylation of glycerol and nucleobases using (Fe,Ni)_3_P or the phosphide substitute Fe_3_P, in mild basic water-base solutions [62,65]. Such conditions could have existed on early Earth, generated by the accumulation of ammonia in localised ponds.

The sugars ribose and glucose, and glycerol, have been the primary organic molecules studied regarding the formation of early organophosphate molecules. The importance of these molecules in today’s cellular energetic metabolism or as structural components of nucleic acid nearly justifies their monopoly in this field. However, these sugars are not as chemically stable as inositol. Both the aldohexose glucose and aldopentose ribose have preferred cyclic hemiacetal structures in water. However, they cycle between this closed conformation and an open chain aldehyde form. Inositol, being a cyclitol, does not cycle between open and closed forms, or possess any reactive aldehyde groups as glucose (Figure 1B,C) and is, thus, more stable. The absence of the aldehyde groups also make inositol “homogenously reactive”: it contains only six equivalently reactive hydroxyl groups. 

The two reports indicating the abiotic synthesis of inositol [56,60] did not investigate which of the nine possible isomeric forms of inositol was detected. It would be interesting to determine if only one, or several, inositol isomers are formed through the formose reaction: thermodynamic stability of inositol phosphate would be affected by its inositol isomeric species. In today’s biology the vast majority of inositol is *myo-*inositol (Figure 1C), though very minor amounts of other isomers are also present. Interestingly, *myo-*inositol has an axis of symmetry between carbons two and five, making this isomer optically inactive.

In eukaryotic cells today, phosphates are added to specific hydroxyls of the inositol ring by position-specific kinases [66,67,68]. Prebiotically, however, following initial equal reactivity, one can envisage that certain abiotic phosphorylation patterns would be preferred over others; for example, steric constraints suggest that phosphorylation of adjacent hydroxyl groups would be unfavourable compared to phosphorylation of diametrically opposite hydroxyl groups. However, it is worth noting that steric constraints do not prevent the synthesis of the fully-phosphorylated inositol ring of IP_6_ in today’s cells [29,30]. IP_6_ (Figure 2C) is by far the most abundant inositol phosphate species present in eukaryotic cells, reaching concentrations from 50 µM in mammalian cells to 0.5 mM in the amoeba *Dictyostelium discoideum*, and it also accumulates in plant seeds [69,70]. Once formed, inositol phosphates possess an extraordinary chemical stability. Boiling for one hour in 1 M hydrochloric or perchloric acid does not degrade IP_6_; while any of the six possible isomers of IP_5_ are merely converted to 2–3 specific IP_5_ species [70]. The isomerization of IP_5_ to another IP_5_ occurs through the process of phosphate jumping (further described in the section below) to adjacent free hydroxyl groups [71]. This occurs at low pH and high temperature. Thus, once prebiotically formed an IP_2_ or IP_4_ species, for example, would likely be isomerised to a more thermodynamically stable subset of IP_2_ or IP_4_ isomers. This suggests that while the six hydroxyl groups might have equivalent chemical reactivity, initially, once formed, the differentially-phosphorylated inositol phosphates (IP_1-2-3-4-5_) will assume a specific subset of isomers in an acidic hot ocean/pond, and not the full array of 64 species that the combinatorial attachment of phosphate groups theoretically predicts.

Therefore, the formation of early prebiotic organophosphate molecules based on inositol offers several advantages over glucose or ribose: inositol does not convert to an open form and, thus, also does not isomerise between stereoisomers as glucose and ribose do; it does not possess a highly-reactive aldehyde group and is, therefore, more stable; it offers up to six equally reactive hydroxyl groups for modification; certain inositol phosphate species synthesized will be favoured or converted to others due to thermodynamic pressure.

### 4.2. The Role of the Iron, Magnesium, and Calcium-Chelating Properties of Inositol Phosphate

Iron meteorites containing schreibersite could have supplied phosphate for the synthesis of the early organophosphate molecules (see above). However, this same mineral may release other reactive ions, particularly iron, along with the phosphide. Iron in water cycles between +2 and +3 states, catalysing the formation of highly-reactive hydroxyl radicals through the Fenton reaction; it must, therefore, be sequestered to avoid unwanted side reactions. Several inositol phosphates, including IP_6_, have the ability to form complexes with iron [72]. Metal complexation studies have indicated that any inositol phosphate with position 1, 2, and 3 phosphorylated can chelate iron [28,73]. Consistent with this, a consensus has emerged that inositol trisphosphate I(1,2,3)P_3_ has the physiological role of complexing iron in today’s cells. While the high reactivity of radicals could increase the repertoire of molecules present in a prebiotic ocean, a controlled radical formation is certainly advisable for a structured chemical evolution. Therefore, the early formation of inositol trisphosphate could have helped to control the iron released from meteoritic mineral schreibersite. The selective synthesis of I(1,2,3)P_3_ is unlikely to have occurred in prebiotic times as it is sterically unfavourable. However, as mentioned above, an interesting chemical feature of inositol phosphates is the ability of their phosphate groups to jump between hydroxyl groups when subjected to low pH and high temperature, plausible on early Earth. This has been exploited experimentally to convert one isomer of inositol phosphate to another [40,74,75,76]. Phosphate jumping between the cis positions 1, 2, and 3 is favoured over phosphate movement to the trans positions 4 and 6 (Figure 1C). Furthermore, the chelating metal may contribute to the formation of the most thermodynamically-stable inositol phosphate isomer. Thus, iron, through an energetically favourable chelation mechanism, might represent the driving force that promoted the formation of specific inositol phosphate isomers. Thermodynamic stability would ultimately determine the equilibrium of inositol phosphate species formed. However, phosphate jumping between the cis positions of the ring should facilitate the prebiotic phosphorylation of the three key positions for the iron-chelating ability of inositol phosphates [28].

The ability of inositol phosphates to form biologically-relevant complexes with bivalent or trivalent cations has been extensively studied, since they must exist in our cells complexed with specific cations [31,77,78]. Under conditions present in today’s cells, IP_6_ (Figure 2B) forms a neutral penta-magnesium species that remains soluble until a remarkably high ~50 µM concentration, while IP_6_/calcium complexes are highly insoluble. The unexpected solubility of IP6/magnesium complexes [31] suggests that, prebiotically, different species of inositol phosphates with differential cation affinities and complex solubilities might have helped discriminate between bivalent cations, for example, eliminating calcium by co-precipitating it with IP_6_, while leaving magnesium in solution by forming relatively soluble complexes.

### 4.3. Inositol Pyrophosphate as the Initial Phosphorylating Agent

The presence of an energetic phosphoanhydride bond in the inositol pyrophosphates (Figure 2C) led, upon their original discovery, to the prediction of the ability of inositol pyrophosphates to participate in a phosphotransfer reaction [79]. The standard free energy of hydrolysis of the pyrophosphate bond in IP_7_ has been estimated to be 6.6 kcal/mol, higher than that of adenosine 5′-diphosphate (ADP) (6.4 kcal/mol) and lower than that of adenosine 5′-triphosphate (ATP) (7.3 kcal/mol) [79]. However, theoretical computation studies attribute a high phosphorylation potential to IP_7_ due to the sterically- and electronically-packed environment of the pyrophosphate moiety [80]. This suggests that if inositol pyrophosphates could have been synthesized prebiotically, they could represent a class of phosphorylating agents for the origin of trans-phosphorylation events.

Two classes of evolutionarily-conserved enzymes synthesise inositol pyrophosphate in eukaryotic cells today [81,82]. However, they can also be synthesised non-enzymatically. Pyrophosphate bonds are synthesized when IP_6_ is freeze dried in the presence of high-energy phosphates such as pyrophosphate, ATP, or creatine phosphate under acidic conditions [83]. Performing this procedure on a positively-charged surface, such as the polycationic beads of a Q-sepharose resin, dramatically increases the yield of inositol pyrophosphate production to about 50% of input IP_6_ [84]. While pyrophosphate, and even inorganic polyP, existed prebiotically [49], the initial activated phosphorus form in that time was most likely phosphide [63]. Therefore, it would be interesting to design protocols using phosphide and a positively-charged mineral as absorbing catalyser to test the possibility of a prebiotic synthesis of inositol pyrophosphates. Nevertheless, evaporation of early Earth acidic ponds might have been able, in the presence of pyrophosphate, to generate inositol pyrophosphates [83]. Thus, it is tantalising to propose that inositol pyrophosphates might have contributed to the origin of a primitive energetic metabolism by participating in phosphotransfer reactions.

## 5. Conclusions

Much of the attention in origin of life research has been given to RNA and DNA: their prebiotic synthesis, condensation, and replication, and to their ability to initiate some kind of enzymology in the RNA world [85]. Attention has also been given to amino acid synthesis and early peptide formation [86], or to the identification of prebiotic ways to generate metabolic networks that recapitulate glycolysis and the citric acid cycle [87]. Often these studies use targeted approaches, such as mass spectrometry focusing on the above-mentioned molecules or their precursors. Thus, the existence of only two reports indicating the prebiotic synthesis of inositol certainly should not be seen as negative. It is very possible, given the metabolic stability of inositol, that this sugar-like molecule, or sugar alcohol, is indeed formed, but has not been looked for in experiments of abiotic chemical synthesis. 

Looking at the intimate relationship between phosphate and inositol in existing cells, the proposed roles for inositol in prebiotic chemistry, namely the initial formation of organophosphate molecules in the form of inositol phosphates, and the differential ions’ selectivity, appear rational. The proposed arguments here must stimulate direct experimental approaches to demonstrate this. It must also be theoretically and experimentally determined which are the most thermodynamically stable inositol phosphates, since it is these species that would have been present on early Earth. Furthermore, I hope that the arguments presented stimulate researchers in the origin of life to consider, besides polyP or pyrophosphate, the inositol pyrophosphates as a possibility for the first molecule with phosphotransfer ability that a primitive metabolism could have utilised.

Finally, I would like to point out that the molecular architecture, i.e., the stereochemical arrangement of phosphate groups around the inositol ring is, in today’s cells, recognised by specific protein domains [33]. This is the accepted mechanism of action for many inositol phosphates. This lets me wonder if the phosphate groups around the inositol ring have created a unique distribution of negative charges important to seed the prebiotic event of molecular recognition [88,89]. However, debating on this possibility is perhaps too speculative and definitely beyond the pay scale of the author.

## Figures and Tables

**Figure 1 life-07-00024-f001:**
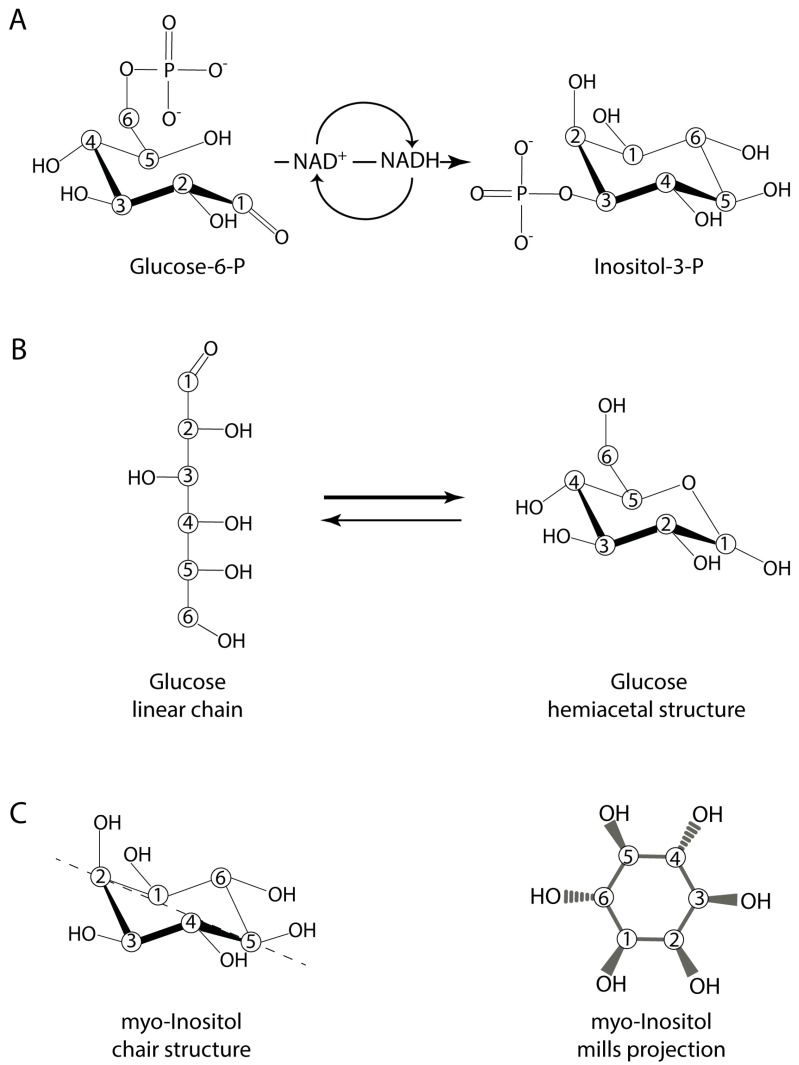
Inisitol and glucose structures. Depicted in (**A**), the enzymatic synthesis of inositol through the action of the inositol phosphate synthase (IPS), an NAD^+^-requiring oxidoreductase that catalyses the isomerization of glucose-6-P to inositol-3-P [26]. In water, the linear chain of glucose (**B**, **left**) is in equilibrium with the hemiacetal cyclic chair structure (**B**, **right**) that, although favoured, is rapidly converted to the open form. The structure of *myo*-inositol, the most common isomer of inositol is depicted in (**C**). The commonly-diagrammed chair configuration (**C, left**) reveals the hydroxyl group at carbon 2 as perpendicular (axial) to the ring plane, while the remaining hydroxyl groups are parallel (equatorial) to the plane of the ring. The dashed line passing between carbons 2 and 5 represent the axis of symmetry of *myo*-inositol. The *myo*-inositol Mills projection is shown in (**C, right**), where the trans 4,6 positions and the cis 1,2,3 positions are evident. Carbons are represented by white circles with numbers indicating their position. The hydrogen of the fourth valence of carbon is omitted for clarity.

**Figure 2 life-07-00024-f002:**
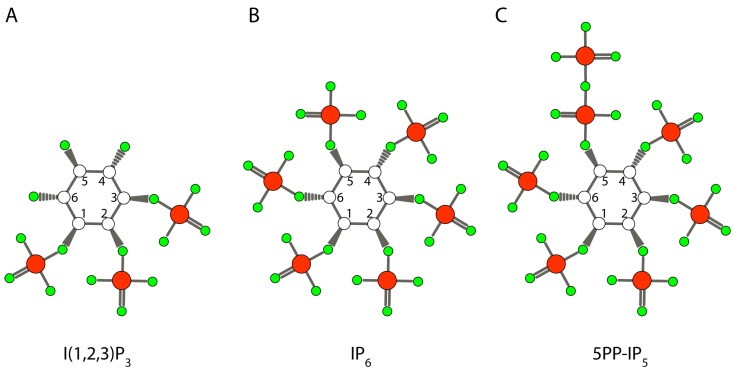
Structure of representative inisitol phosphates. Inositol phosphates represent a complex family of molecules present in eukaryotic cell cytosol [19,27]. More than forty different inositol phosphates have been identified. In (**A**) the structure of I(1,2,3)P_3_ is represented. Phosphorylation of these three positions confers the ability to bind iron [28]. In (**B**) the structure of the fully-phosphorylated ring of IP_6_ (also known as phytic acid) is represented. IP_6_ is particularly abundant in plant seeds and is the most abundant inositol phosphate present in eukaryote cells [29,30]. IP_6_ binds with high affinity to magnesium, forming a soluble pentamagnesium complex in the cytosol [31]. It is the main, but not exclusive, precursor of the energetic inositol pyrophosphates [24]. Shown in (C) is the structure of the prototypical inositol pyrophosphate IP_7_, so named as it contains seven phosphate groups. Specifically the isomer 5PP-IP_5_, where the pyrophosphate is in position 5, is represented. Carbon is represented with white circles, phosphorus with red circles, and oxygen with green circles.

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
