# Peer review of "Has Inositol Played Any Role in the Origin of Life?"

_life, 2017, doi:10.3390/life7020024_

Round 1

Reviewer 1 Report

The paper "Has inositol played any role in the origin of life?" presents a case that a sugar alcohol, inositol, could have served as a useful substrate for prebiotic phosphorylation and synthesis.

The mark of a good paper is that it is one that teaches something.  In this case, the paper presents an informed argument on the chemistry of inositol phosphates, which are prevalent in life and which play several key roles in eukaryotes.  Such molecules have been generally ignored in prebiotic chemistry, principally because they are not the sugars that play direct roles in RNA synthesis. However, there are several advantages to considering inositol phosphates as prebiotic reagents, including their ability to store phosphate, serve as signal molecules (as informational molecules!), phorphorylate reagents, and store energy with pyrophosphate bonds.  

The concerns with this paper are relatively minor.  I would alter some of the language, especially in light of the paper by Goldford et al. (2017).  This paper does present an alternative view of phosphate-centric metabolism, and even if not correct, addition of this material doesn't hurt the current paper any:

Line 10: Change "must" to "may"

Line 37: "Must" to "should"

Line 38: The Goldford paper is relevant here and could be cited.

Line 41: Trisulfate metabolisms aren't generally argued for.  Thioester metabolisms are more reasonable.  Change the text appropriately.

The other section of concern is on the prebiotic chemical pathway to inositol (Section 3).  It is unclear why nitrogen compounds would need to play any role (HCN/formamide), given the lack of N in inositol.  Prebiotic pathways to inositol more likely include formose-style reactions with disproportionation driving alcohol formation with carboxylic acid formation, or by reducing agents acting on the formose mixture (e.g., Ricardo et al. 2004).  For instance, Shigemasa's work in the 1980s show sugar alcohols to be significant constituents in many of their experiments. Cooper's (2001) analysis of meteorites show that glycerol is more abundant than the equivalent 3-carbon sugars, justifying this assumption.  Perhaps leaving this section more open, but arguing that formation of 6-carbon sugars should be accompanied by 6-carbon sugar alcohols would be reasonable.

Uncapitalize "schreibersite" throughout, as the mineral name is not a proper noun. 

The formation of inositol phosphate is certainly reasonable for many of the reasons outlined in this paper.  Its lack of study is indeed due to both a lack of attention, and due to the potential for complexity in its analysis.  But as a prebiotic parallel, glycerol (an easier sugar alcohol to study than inositol) is quite easy to phosphorylate under prebiotic conditions (as shown in the paper by reference 7, and also by Gull et al. (2014), and by Gull et al. (2017, in review at https://www.preprints.org/manuscript/201704.0168/v1  ).  These do highlight potential for making inositol-phosphates, if phosphorylation can be coupled to prebiotic inositol synthesis.

References

Goldford, J. E., Hartman, H., Smith, T. F., & Segrè, D. (2017). Remnants of an ancient metabolism without phosphate. Cell168(6), 1126-1134.

Gull, M., Zhou, M., Fernández, F. M., & Pasek, M. A. (2014). Prebiotic phosphate ester syntheses in a deep eutectic solvent. Journal of molecular evolution78(2), 109-117.

Ricardo, A., Carrigan, M. A., Olcott, A. N., & Benner, S. A. (2004). Borate minerals stabilize ribose. Science303(5655), 196.

Shigemasa, Y., Kawahara, M., Sakazawa, C., Nakashima, R., & Matsuura, T. (1980). Formose reactions: IX. Selective formation of branched sugar alcohols in a modified formose reaction and factors affecting the selectivity. Journal of Catalysis62(1), 107-116.

Author Response

Reviewer 1’s comments are remarkably nice: they stated that the manuscript “teaches something”. This reviewer just listed a series of minor corrections. The text has been modified on Lines 10, 37, 38 as indicated by the reviewer and the mineral name "schreibersite" has been un-capitalized throughout the manuscript. Furthermore, additional references regarding formose reaction and glycerol phosphorylation have now been included in the revised version of this work.

The only suggestion not taken forward by the author is on Line 41, where the reviewer asked for Trisulfate to be changed to Thioester. The author understands and fully agrees with the reviewer’s point. Thioesters are very important energetic bonds for today’s cellular biochemistry, while sulphates are not. However, the author in line 41 aims to contrast the phosphate-based bioenergetics that evolved to ‘ATP’ with sulphur-based bioenergetics that did not evolve to ‘ATS’ (a hypothetical adenosine tri-sulphate). In this context, the use of trisulfate better matches ATP and sounds better than ATT (adenosine tri-thioester).

The reviewer questioned that since N (nitrogen) is not present in the structure of inositol, why would nitrogen compounds (HCN/formamide) play any role in inositol prebiotic synthesis. The reviewer indicated that formose-style reactions, starting from formaldehyde, are likely responsible for inositol prebiotic synthesis. The author fully agrees. Cyanide and formamide are plausible prebiotic small molecules, from which formaldehyde is generated. Thus the author discusses these nitrogen-containing molecules mainly as formaldehyde precursors. The revised version of the manuscript has been edited to simplify and clarify this issue. The misleading sentence ‘starting from the simple building block of cyanide (HCN)’ is now deleted.

Reviewer 2 Report

The article “Has inositol played any role in the origin of life?” proposes that inositol might have functioned as a phosphate storage molecule, cofactor, and/or transphosphorylation agent during the earliest stages of life. The article is well-written, and provides some compelling arguments that might prompt others to examine inositol in context of prebiotic chemistry. I recommend accepting this article for publication. A few additional comments: 1. Most compelling to this reviewer is that inositol is both found in model prebiotic reactions that form sugars, while being more stable than sugars. As the author points out, this might have led to the buildup of inositol, thus the molecule would have been more available for incorporation into early biological functions. An easy experiment to try (outside of the scope of the article under review) would be show a phosphorylation completion between inositol and aldose and ketose sugars. Heating alcohols with a phosphate source and urea to dryness or in an organic solvent (e.g., formamide) is generally used to form organophosphates. Sugars, however, degrade at temperatures typically required for phosphorylation (ca. 100 C). 2. In section 4.2 there is a discussion of the unfavorable formation of I(123)P3. The authors are correct to assume that transesterification will lead to the most energetically favorable molecules dominating. While I(123)P3 might be sterically unflavored, the chelation of iron or other metals would stabilize this isomer, therefore a driving force exists to promote the formation of I(123)P3 via energetically favorable chelation. The authors might want to discuss this point further. 3. On page 12, line 331, inositol is called a sugar, it would be more appropriate to call the molecule a sugar analog. 4. A few grammatical issues exist in the following lines. 84, 207, 242, 272/273

Author Response

The author is pleased by the lovely comments of reviewer 2. The author is delighted to hear he succeeded in his main objective to “provide[s] some compelling arguments that might prompt others to examine inositol in context of prebiotic chemistry.”

They recognised that the metabolic stability of inositol might have played a favourable role in early biological functions, and suggests comparative (between inositol and other sugars) phosphorylation experiments that the author is planning in the near future.

They agree with the author that phosphate jumping between hydroxyl groups will lead to the formation of the most energetically favourable molecules. However, they intelligently point out that the chelating metal associated with inositol phosphates might also contribute to the formation of the most thermodynamically stable isomeric molecules. They proposed “energetically favorable chelation” as the driving force to promote the formation of specific inositol phosphate isomers. This attractive hypothesis is discussed in the revised version of the manuscript.

The reviewer suggested calling inositol a “sugar analog” instead of sugar. The author agrees to this slight change of nomenclature and on page 12, line 331, inositol is now defined as sugar-like or sugar alcohol.

The few grammatical issues (lines. 84, 207, 242, 272/273) indicated by the reviewer have been corrected.

Reviewer 3 Report

The paper deals with new perspectives in the phosphorylation of organic compounds from inositol and the inositol phosphates. This approach is very interesting and seems to be realistic. Only few paper are interested to this approach and this perspective work allows to imagine new relevant experiments. I recommande its publication without modification. 

Author Response

The author thanks this reviewer for their positive views. They considered the proposed ideas as “very interesting” and “realistic” and recognises that “this perspective work allows to imagine new relevant experiments”. The author is particularly pleased by this comment, since this was the main objective of the author in writing this article.